# Tissue-Resident NK Cells: Development, Maturation, and Clinical Relevance

**DOI:** 10.3390/cancers12061553

**Published:** 2020-06-12

**Authors:** Elaheh Hashemi, Subramaniam Malarkannan

**Affiliations:** 1Laboratory of Molecular Immunology and Immunotherapy, Blood Research Institute, Versiti, Milwaukee, WI 53226, USA; ehashemi@versiti.org; 2Department of Microbiology and Immunology, Medical College of Wisconsin, Milwaukee, WI 53226, USA; 3Department of Medicine, Medical College of Wisconsin, Milwaukee, WI 53226, USA; 4Department of Pediatrics, Medical College of Wisconsin, Milwaukee, WI 53226, USA

**Keywords:** NK cells, tissue-resident, anti-cancer responses

## Abstract

Natural killer (NK) cells belong to type 1 innate lymphoid cells (ILC1) and are essential in killing infected or transformed cells. NK cells mediate their effector functions using non-clonotypic germ-line-encoded activation receptors. The utilization of non-polymorphic and conserved activating receptors promoted the conceptual dogma that NK cells are homogeneous with limited but focused immune functions. However, emerging studies reveal that NK cells are highly heterogeneous with divergent immune functions. A distinct combination of several activation and inhibitory receptors form a diverse array of NK cell subsets in both humans and mice. Importantly, one of the central factors that determine NK cell heterogeneity and their divergent functions is their tissue residency. Decades of studies provided strong support that NK cells develop in the bone marrow. However, evolving evidence supports the notion that NK cells also develop and differentiate in tissues. Here, we summarize the molecular basis, phenotypic signatures, and functions of tissue-resident NK cells and compare them with conventional NK cells.

## 1. Introduction

Immune cells primarily reside in the lymphoid organs and traffic to the sites of infection or tumor transformation [1,2,3,4]. At the site of pathology, the migrated lymphocytes initiate responses to eliminate the threat. In contrast to this dogma, recent studies reveal the localized development and functions of tissue-resident immune cells [5,6,7,8,9,10]. Natural killer (NK) cells are the major innate lymphocytes [11,12,13,14]. NK cells mediate cytotoxicity [15,16,17,18,19] and produce significant quantities of inflammatory cytokines, including interferon-γ (IFN-γ) [20,21,22,23,24] and tumor necrosis factor-alpha (TNF-α) [25,26,27,28]. Our knowledge about the molecular basis of the development and functions of NK cells is progressing rapidly. Earlier studies have established that human NK cells consist of two major subsets (CD56^bright^ and CD56^dim^) [29,30,31,32]. Recently, we and others, using single-cell RNA-based transcriptomic analyses, have shown that human NK cells are highly heterogeneous with diverse functions [8,33]. Most of our knowledge about NK cells are obtained using mouse spleen or bone marrow (BM), and human peripheral blood mononuclear cells (PBMC). These cells are referred to as conventional NK cells (cNK). Recent studies have started shedding insights into less-studied tissue-resident NK cells (trNK) [8,34,35]. The dichotomy between the cNK and trNK is the focus of this review.

NK cells lack clonotypic receptors and mediate their effector functions without prior sensitization [36]. They exert their function with germline coding activating and inhibitory receptors [37]. Activating receptors such as NKG2D and NCRs are expressed in both cNK and trNK, and recognize induced self-ligands or antigens from pathogens [38,39,40,41,42,43,44,45,46]. Induced self-ligands include non-classical major histocompatibility complex class I [46] such as human MIC-A, MIC-B [39,47,48,49,50,51], and murine H60 [45,52,53], Rae (α-ε) [54], and Mult-1 [55,56]. Multiple viral proteins, including hemagglutinin (HA), are recognized by NCRs [1,2]. It is important to note that the expression of ligands for both NKG2D or NCR1 is predominantly occurs on epithelial cells, endothelial cells, and monocytes that are infected or stressed [3,4]. These observations provide a conceptual framework of how both cNK and trNK cells can be activated by similar mechanisms. Based on the expression of CD16 (FCRγIII), human cNK cells can be further divided into two subsets, CD56^bright^CD16^−^ and CD56^dim^CD16^+^ [57,58,59]. The distribution of these subsets among the cNK and trNK differs significantly. Inhibitory receptors such as human killer immunoglobulin-like inhibitory receptors (KIR) [60] and murine Ly49s [61] primarily recognize classical Major Histocompatibility Complex Class I (MHC-I) and form the basis for ‘licensing’ that allows NK cells to differentiate between ‘self’ and ‘non-self’ [5,6,7,8,9]. Notably, cNK and trNK cells express differing levels or types of KIRs, emphasizing the potential divergent effector functions of these subsets.

cNK cells kill infected or transformed cells that have null or low expression levels of MHC-I molecules [62,63]. Individuals lacking NK cells are prone to viral infections [64,65,66,67]. cNK cells recruit innate cells, such as dendritic cells (DCs) [68], neutrophils, and macrophages, to initiate and augment immune responses [69,70]. During this interaction, naïve cNK cells are primed to augment their proliferative and functional capabilities. DCs produce a vast array of cytokines, including IL-15, IL-12, IL-23, IL-27, and IL-18, that have a direct role in the development and functions of cNK cells [71]. Interactions between DCs and cNK cells are essential for this priming [72] and to regulate adaptive immune responses [73,74]. Activation of DCs with Type-1 IFN-α/IFN-β by cNK cells results in the production and trans-presentation of IL-15/IL-15Rα complexes on the cell surface of plasmacytoid DCs [72]. The stimulation of DCs with Type-1 IFN-α/IFN-β by cNK cells, resulting in the trans-presentation of IL-15/IL-15Rα from DCs to the IL-15Rα/IL-2Rβ/IL-2Rγ complex on cNK cells, represents one of the necessary transition steps from innate to adaptive immune responses [75,76]. IL-12 produced by DCs prime cNK cells to produced Type 1 interferons, which in turn help to mature DCs and help in CD4+ T helper-1 (Th1) T cell priming in lymph nodes (LNs) [77]. Thus, the interplay between cNK cells and DCs form an essential link for both cNK and T-cell mediated immune responses [78]. Apart from the DCs, several other tissue-resident myeloid and stromal cells produce IL-7, IL-15, and IL-21. The contributions of these cells towards the development, tissue-retention, and functions of trNK cells are not fully understood. cNK and trNK are two developmentally and functionally divergent subgroups of NK cells. Here, we discuss the ontology, phenotypic and functional characteristics of the trNK cells.

## 2. cNK Versus trNK Cells

cNK cells develop in the BM [79]. Hematopoietic stem cells (HSCs) in the BM commit to early lymphoid progenitors (ELP) that express high levels of cKit, Sca1, and Flt3 [80,81]. ELPs develop into common lymphoid progenitors (CLPs) that possess a decreased expression of Sca1 and a high level of interleukin (IL)-7 receptor [82,83]. Multiple cytokines play essential roles during the development of cNK cells [79,84]. IL-15 is one of these which uses the common receptor gamma chain (γc) [85,86]. The expression of IL-2β receptor (CD22) results in an irreversible lineage commitment to cNK cells [83,84,87]. Immature cNK cells express CD117 and integrin α_2_ (DX5) [88]. In the following stages, these cells go through CD27^+^CD11b^−^ to CD27^+^CD11b^+^ to CD27^−^CD11b^+^ [89]. Most of the cNK cells trafficked from the BM into the periphery predominantly are CD27^−^CD11b^+^ [79,84].

In contrast to the cNK cells, the developmental stages of trNK cells are poorly defined. Whether there are distinct precursors for trNK cells is yet to be elucidated. At what stage the commitment to trNK cells happen, is unknown. Moreover, whether specific cytokines drive NK cell progenitors (NKPs) toward tissue residency is an open question (Figure 1). Cytokine-mediated metabolic reprogramming plays an essential role in the developmental progression of cNK cells. In this context, the qualitative roles of cytokines on trNK cells are yet to be established. Cytokines like IL-12 and IL-15 induce different levels of the expression of nutrient transporters in cNK cells [90]. It is important to note that the CD56^bright^ CXCR6^+^ trNK cells express lower levels of Glut1, but higher levels of the amino acid transporter CD98 compared to CD56^bright^ NK cells from peripheral blood following stimulation [90]. The functional relevance of such differences requires further investigation.

Cytokine profiles of trNK cells are different compared to that of cNK cells. Effector functions of cNK cells include mediating natural cytotoxicity and producing inflammatory cytokines such as TNF-α, IFN-γ, GM-CSF, IL-5, IL-8, IL-10, and IL-13 [91,92,93,94]. cNK cells also produce multiple chemokines, including XCL1, XCL2, CCL1, CCL3, CCL4, CCL5, CCL22, and CXCL8 [33,95,96,97], which help to generate and sustain the inflammatory environment [98,99]. Compared to cNK cells, the liver trNK cells have a higher expression of TNF-α and GM-CSF [8]. Moreover, a significant proportion of liver-resident NK cells are double-positive for IFN-α and TNF-α [8]. TrNK cells are lineage distinct from cNK cells with different requirements of transcription factors (Figure 2). For example, cNK cells were absent in Nfil3 (nuclear factor IL-3-regulated protein)-deficient mice, while these mice possess trNK cells in the liver, skin, and uterus [8,100]. Moreover, NK cells in mouse salivary glands develop in the absence of Nfil3 [101]. cNK cell development is independent of thymic influence, but the thymic-reliant NK cells are absent in nude and GATA3-deficient mice [102]. NK cells are also found in mucosal tissue, including the small intestine. Intestinal Lin^−^NKp46^+^NK1.1^+^ cells consist of Eomes^+^RORγt-fate map^−^ cNK cells, which express CD27 and lack IL-7Rα, along with a smaller fraction of Eomes^−^RORγt-fate map^+^ NK cells [103] that is similar to liver-resident NK cells [104]. Thus, cNK and trNK cells significantly differ in their requirement of the developmental niche and transcriptional networks [105,106]. The topological organization of trNK cells within the microenvironment in organs such as lung or uterus holds clues about their functions, interacting partners, and cellular signals [107,108,109,110]. 

Human cNK cells consist of two major subsets, CD56^bright^ and CD56^dim^. Functionally, CD56^bright^ NK cells have an increased capacity of cytokine production compared to CD56^dim^ NK cells, which are cytotoxic [111,112]. In peripheral blood, the majority of cNK cells are CD56^dim^, while CD56^bright^ is abundant in lymphoid tissues and outnumber in LNs, endometrium, and decidua [34,113,114,115,116]. Although cNK and trNK cells share common effector functions, recent studies indicate that the tissue site can dictate the potential functions of trNK subsets (Figure 1). Recent studies demonstrate that the trNK cells in the thymus, liver, lymph nodes, and uterus follow developmentally distinct pathways. For example, Dogra P et al. examined the expression of GzmB in CD56^dim^CD16^+^ compared to CD56^bright^CD16^−^ NK cells across blood and lymphoid tissues [117]. They found that the frequency of GzmB^+^ in CD56^dim^CD16^+^ NK cells is significantly higher in the lung compared to LNs and gut [117]. They also found that the expression of FcεRIγ depends on tissue type. Tonsils, gut, and LN CD56^dim^CD16^+^ NK cells express substantially higher levels of FcεRIγ compared to CD56^dim^CD16^+^ NK cells in the blood, BM, spleen, and lungs [117]. Moreover, trNK cells have specific functions that are exclusively related to organ-specific niches. For instance, trNK cells in the uterus have a role in placental vascular remodeling [118,119], fetal growth [120], and memory of pregnancy [121]. Irrespective of these advances, the developmental origin, subset heterogeneity, functional uniqueness of the trNK cells are yet to be fully defined. In the following sections, we focus on the specific features of trNK cells in lymphoid and non-lymphoid organs. 

## 3. Thymic NK Cells

The presence of NK cells in the thymus has been known for a long time [122], and this subset is distinct from the cNK cells [102]. In mice, thymic NK cells are defined by CD127^+^Ly49^low^CD11b^low^, which represents an immature phenotype [8,102]. CLPs that migrated from the BM into the thymus can commit to becoming T/NK progenitors, and most of them develop into CD4^−^CD8^−^ double negative (DN) precursors [123,124,125,126,127]. These CD4^−^CD8^−^ thymocytes contain a limited number of precursors with the potential to commit to NK lineage [128]. This highly heterogeneous DN1 (c-Kit^high^CD44^+^CD25^−^CD122^−^NK1.1^−^) thymocytes that are yet to initiate the T cell receptor (TCR)-beta chain rearrangement were able to develop into NK1.1^+^ NK cells following in vitro culture with IL-15, IL-7, Flt3L, and stem cell factor (SCF) [128]. These cells show immature BM-derived NK features and expressed CD127 [102]. In addition to DN1, another transitional subset of DN2 (DN2a/DN2b; CD44^+^CD25^+^) also has the potential to commit to NK lineage [129]. 

The thymic NK cells also traffic to other tissues. Among total NK cells, CD127^+^ thymus-derived NK cells are present in around 5% in the BM, liver, and spleen; however, they are enriched (15–30%) in the mesenteric lymph nodes (LNs). These cells are phenotypically similar to thymic CD127^+^ NK cells [88]. Like thymus-resident NK cells, they express lower levels of Ly49A, C/I, and G2 and lack the expression of Ly49D [102]. However, this notion of thymic-origin trNK cells present in the LN is complicated by the fact that there are early progenitors present within the LN. Thus, whether the thymic precursor migrates to LN to further develop or mature in the thymus and traffic to LNs is not known. Functions of thymic NK cells are largely not known. Unlike immature NK cells, thymic NK cells are functionally active, and they can produce cytokines, including IFN-γ [102]. The predicted functions include the formation and maintenance of thymic architecture, regulation of thymopoiesis, elimination of transformed thymocytes, and maintenance of T cell clonal diversity. Thymic NK cells may influence the T cell ontogeny through diverse mechanisms, including the elimination of negatively selected thymocytes using perforin/granzyme, Fas/FasL/TNF-related apoptosis-inducing ligand (TRAIL), or through the production of cytokines and chemokines. Similar to cNKs, thymic NK cells require Nfil3, which regulates the transcription of Id2 [130]. However, in contrast to cNK, thymic NK cells develop optimally with minimal phenotypic changes in the absence of Id2 [131]. Upstream E-box transcription factor, Ets1 regulates Id2; and Ets1 is required for thymic NK cell development [132]. This could be due to the reduced reliance of thymic NK cells on IL-15, which turns on Id2-dependent developmental progression [100,133]. Since thymic NK cells primarily depend on IL-7 and not IL-15 in the niche within the corticomedullary junction (ETP/DN1) or cortex (DN2a/DN2b) of the thymus, it is plausible that they utilize an alternative transcriptional program. The thymic NK cells require GATA3 for their development [102,128]. Since GATA3 regulates IL-7Rα receptor expression in a Notch-dependent manner, the absence of this transcription factor results in the lack of thymic NK cells [134,135]. This is further supported by the fact that both GATA3-deficient and athymic nude (*Foxn1^−/−^*) mice lack the thymic NK cells [102]. Most of the characterizations of thymic NK cells are based on murine models. One potential human subset that may represent the murine thymic counterpart is human LN-resident CD34^dim^CD56^bright^CD16^−^CD127^+^c-Kit^high^ NK cells. Detailed future studies are required to fully appreciate the phenotypic and functional uniqueness of human thymic NK cells.

## 4. Liver-Resident NK Cells

NK cells were first described as ‘pit cells’ in the rat liver in the 1970s [136]. NK cells in the human liver represent up to 30–50% of all hepatic lymphocytes in contrast to 5–16% in PBMC [137]. Given the fact that the liver represents the largest internal organ by mass, the number of trNK cells is significant, and thereby their functions [138]. The link between the liver and the NK cells starts in the fetal liver, where the earliest hematopoiesis occurs [139]. The adult BM is the major generative organ in later life [140,141]. Moreover, the hepatic vascular system has unique characteristics relative to other organs [138]. Two afferent vessels come into the liver, hepatic artery, and portal vein [138]. Terminal branches of the hepatic portal vein and artery mix as they enter the sinusoids [142]. trNK cells are found in the sinusoidal lumen and are associated with sinusoidal endothelial and Kupffer cells, the liver-resident macrophages [34,136,143,144]. 

In human and mouse liver, NK cells consist of heterogeneous populations [8]. Recent studies helped to identify the phenotypic and transcriptomic differences between the circulating cNK and liver-resident NK cells [8,90,145]. In mice livers, there are two populations of NK cells, distinguished by mutually exclusive expression of CD49a and DX5 [145]. CD49a^−^DX5^+^ is similar to splenic cNK, whereas CD49a^+^DX5^−^ are trNK cells [8,145]. In parabiotic mice, the host liver contains CD49a^+^DX5^−^ NK cells of host origin and circulating CD49a^−^DX5^+^ NK cells derived from both host and the other parabiont, indicating that the CD49a^+^DX5^−^ cells are trNK cells [8,145]. In contrast, the CD49a^−^DX5^+^ NK cells are cNK cells [8]. The trNK cells are similar to immature cNK cells because they express the conventional markers NK1.1 and NKp46; but, low levels of CD11b and lack CD49b [8]. CD49a^−^DX5^+^ liver subset displays high levels of TRAIL [8,146]. Sojka et al. show that liver trNK cells develop independently of the thymus or GATA3, which is the required transcription factor for thymic NK cells [8]. Moreover, in Nfil3-deficient mice, a normal number of liver trNK cells were present, while these mice had no cNK cells in the spleen [8,100]. These data suggest CD49a^+^DX5^−^ trNK cells in the liver are of residents and a distinct lineage from CD127^+^ thymic NK and cNK cells [8].

In humans, intrahepatic resident NK cells (ihNK) are imprinted in a liver-specific signature with high cellular heterogeneity [138]. ihNK contained a higher proportion of CD56^bright^ NK cells compared to PBMC. First, CD56^bright^CXCR6^+^ were defined as markers for liver tissue residency, but later among CD56^bright^CXCR6^+^ population, a subset, which are Eomes^hi^Tbet^lo^ are considered as trNK cells that are absent in the blood [147]. CD56^bright^CXCR6^+^ trNK population is characterized by high expression of CD69 and CXCR6 and low expression of DNAM, perforin, and granzyme B, which attributes a non-cytotoxic function for this trNK subset [34,148]. These cells are located primarily in the sinusoids where they express a unique repertoire of chemokine receptors, including CCR5 and CXCR6, and retained in the microenvironment through the interaction of these receptors and their ligands expressed on sinusoidal endothelial cells [34]. Liver-resident NK cells are also characterized by specific transcription factors. One of them, Hobit, is highly expressed in CD56^bright^ trNK cells, along with chemokine receptors, CXCR6, adhesion molecules CD69, and CD49a that is associated with liver residency [149]. Hobit^pos^CD56^bright^ trNK cells in the liver also possessed a higher level of T-bet and Blimp-1 when compared to Hobit^neg^CD56^bright^ trNK cells [149,150]. The effector functions of liver-resident NK cells may be non-cytotoxic, but produce a higher level of inflammatory cytokines (Figure 2 and Figure 3). In patients with cirrhosis, CD49a^+^ NK cells include subsets of CD34^+^CD25^+^ cells that proliferate in response to low doses of IL-2 [151]. This suggests that human liver-resident NK cells have distinct functional characteristics and may contribute to liver inflammation and fibrosis.

## 5. Lung-Resident NK Cells

When the lung alveolar surfaces are exposed to pathogens and harmful materials, the innate immune system, including NK cells, provides initial protection [152,153,154,155,156]. In humans, NK cells accounting for about 10–20% of the lymphocytes in the lungs [107] In mice, 10% of lymphocytes consist of NK cells [107,157]. The majority of NK cells in the human lung are circulating CD56^dim^CD16^+^, and the frequency of mature CD57^+^NKG2D^−^ cells is higher in the lung compared to peripheral blood [7]. These cells express higher levels of KIR2DL1, KIR2DL2/S2, KIR2DL3, and KIR3DL1. In human lung, approximately 10% of CD56^dim^CD16^+^ and 75% of CD56^bright^CD16^+^ NK cells, which also express CD69. CD69 is a hallmark for tissue residency [158,159,160]. Michaelsson et al. first demonstrated the presence of lung trNK cells [161]. They report three distinct lung trNK cell subsets, CD16^−^CD69^+^CD49a^+^CD103^+^, CD16^−^CD69^+^CD49a^+^CD103^−^, and CD69^+^CD16^−^ [161]. The first two subsets express a higher level of ITGA1 (CD49a), ZNF683 (Hobit), RGS1, and RGS2, and lower levels of SELL (CD62L), S1PR5, and KLF3 transcripts compared to the CD69^−^ subset [161]. These have been previously reported as the hallmark genes of CD8^+^ tissue-resident memory T cells (T_RM_) [162,163]. CD69^+^CD16^−^ subset expresses tissue residency markers to a lesser extent; therefore, this subset is considered similar to cNK cells [161]. Comparison of lung trNK to cNK cells or CD8^+^ T_RM_ cells indicates that these are similar to BM NK cells and CD8^+^ T_RM_ cells rather than splenic naïve CD8^+^ T cells [161]. Lung trNK cells play an important role in respiratory diseases, including infectious diseases, allergy, and cancer [107,164].

Circulating cNK cells from the periphery are recruited to the lung during infections. For example, during the early stages of *Mycobacterium tuberculosis* (MTb) infection, NK cells with upregulated CD69, IFN-γ, and perforin accumulate in the lungs [165]. Moreover, NK subsets representing CD94^high^KIR^low^ were recruited from the blood into the lungs during respiratory tract inflammation [166]. In mice, CD11b^high^CD27^low^ cells are the predominant lung-resident NK subset [167] and shown to have a role in the control of pulmonary tumor growth [164]. NK cells in tumor tissues of patients with non-small-cell lung cancer (NSCLC) show distinct receptor expression patterns with lower expression of NKp30, NKp80, KIR2DL1, and KIR2DL2 and higher expression of NKp44, NKG2A, CD69, and HLA-DR [107]. These studies characterize lung trNK cells and their functions, but detailed studies are needed to uncover the developmental progression, transcription factor profiles, and functions of the NK cell subsets in the lung. 

## 6. Lymph Node-Resident NK Cells

LNs provide niches with diverse and concerted interactions of immune cells, which facilitate robust responses [114,168,169,170,171]. The human body has 500 to 600 LNs that are distributed to provide region-specific immune responses [172]. The LN is divided into three central regions, cortex, paracortex, and medulla. The cortex is the outer region of LN that contains B cell follicles and interfollicular zone (IFZ) [173,174,175]. In LNs, NK cells, which constitute 2–5% of lymphocytes [35,113,116,176], are localized in IFZ along with γδ T cells, natural killer T (NKT) cells, and innate-like CD8^+^ T cells [177]. NK cells in the LN consist of multiple subsets, and they are cNK, NK cells that possess the thymic origin, and a unique lymphoid tissue-resident NK (Ltr-NK) that are shown to develop within this organ. Most of these NK cells are located adjacent to lymphatic sinus-lining sentinel macrophages [178]. Stromal cells regulate the movement of lymphocytes, including cNK cells within the LN by secreting chemokines such as CCL21, CCL19, and CXCL13 [179]. 

A recent study using single-cell RNA sequencing suggests that the organization of the LN into distinct functional compartments is due to the specific type of stromal cell subsets present in the cortex, paracortex, and medulla of the LN [179]. T cell-zone reticular cells (TRCs) that express CXCL9 and CXCL10 are also located in IFZ and play an essential role in positioning dendritic cells (DCs) and T cells within the IFZ. However, neither the interaction of LN-NK subsets with other immune cells within IFZ nor the presence of an exclusive LN-NK niche has been explored. In human, more than 75% of all NK cells in LNs are CD56^bright^ [35,116,180]. Circulating CD56^bright^ NK cells enter the LNs via high endothelial venules (HEV) and afferent lymph vessels [31,181], which express CCR7 and CD62L [182,183]. They can be recruited by the CCL19 and CCL21 chemokines, which are highly expressed in LNs [113]. Ltr-NK subset in the LNs consists of 60% of total NK cells and covers the majority of the CD56^bright^ compartment. These Ltr-NK cells were identified based on the co-expression of CD69 and CXCR6 [35]. Ltr-NK cells also express NKp46, and most of them are CD16^−^CD49a^−^CD27^+^ [35]. 

Ltr-NK cells do not express DNAX accessory molecule 1 (DNAM1), an activating receptor that is uniformly expressed on circulating CD56^bright^ NK cells [35]. Functionally, Ltr-NK cells produce less IFN-γ and reduced capacity to lyse K562 cells compared to the circulating CD56^bright^ NK cells [35]. Specifically, CD56^dim^CD16^+^ Ltr-NK cells express substantially higher levels of FcεRIγ compared to CD56^dim^CD16^+^ cNK cells in the blood, BM, spleen, and lungs. Freud et al. identified a new CD34^dim^CD45RA^+^ immature NK subset in the LN, which may develop into Ltr-NK [184]. Recently, Ferber et al. showed that LNs contain CD56^bright^CD127^+^CD161^+^ cells that represent less differentiated precursor NK cells compared to other peripheral sites [117]. Mice lacking Eomes and T-bet failed to develop cNK cells with a modest reduction in Ltr-NK cells [185]. Moreover, it was reported that the Klf4-deficient mice contain lower numbers of cNK cells in the blood and the spleen but normal numbers of trNK cells in other organs such as the liver and LNs [186]. *Tox*^–/–^ mice have a similar phenotype to *Id2*^–/–^ mice and lack mature cNK cells in the periphery as well as LNs [186,187]. Further studies are needed to investigate the possibility of LN as another primary site for the development of a unique Ltr-NK subset. The transcriptional requirement for the development of Ltr-NK cells needs to be determined.

## 7. Uterine NK Cells

Potentially, the most relevant immune cells during pregnancy are NK cells. In mice, during the non-menstrual period, NK cells make up 30% of endometrial lymphocytes, and this number increases significantly (70% of lymphocytes) during pregnancy and decidualization [188,189]. Almost all the NK cells in the non-pregnant uterus (endometrium) and pregnant uterus (decidua) are CD56^bright^ [31,190]. Uterine CD56^bright^ NK cells (uNK) express CD94, NKG2A, and lack of expression of CD16 (Figure 1 and Figure 2) [190]. These uNK cells are distinct from peripheral blood in multiple aspects. Most of the uNK cells express CD49a and CD103 [191]. Moreover, KIR receptors on uNK cells are distinct from cNK cells [192,193]. Decidua NK cells are educated by maternal HLA-C molecules, which influences the expression of KIR receptors [192]. Due to this unique ‘licensing’ program, the expression levels of KIR2DL1^+^ and KIR2DL3/L2/S2^+^ on decidua NK cells are increased compared to cNK cells [192]. uNK cells are transcriptionally distinct from cNK cells, and microRNA profiles of these cells are also different [194]. Specific miRNA, miR-362-5p, is highly upregulated in decidua NK cells, which targets CYLD, a negative regulator of the NF-κB signaling pathway [194]. 

uNK cell frequency may change through the menstrual cycle or during pregnancy. Sojka et al. used *Ncr1-iCre-Rosa^mt/mG^* mice containing membrane-bound Td-tomato to track NK cells during pregnancy [188,195,196,197]. In this mouse, when Cre is expressed, the floxed Td-tomato cassette and the stop codon are removed, and expression of membrane-bound GFP occurs, facilitating the tracking of uNK cells. Using this method, Sojka et al. show that on the gestational day 6.5, the decidua basalis contained proliferating GFP^+^ uNK cells, before the development of mesometrial lymphoid aggregate of pregnancy (MLAp), challenging the idea of considering MLAp as a source of immature uNK cells [198]. At gestational day 10.5 (mid-gestation), they found GFP^+^ uNK cells within the prominent MLAp structure. Shortly after midgestation, the number of GFP^+^ uNK cells began to decline at the implantation site [198]. Remarkably, at 2.5 days post-partum, the number of GFP^+^ uNK cells start to resemble those in the non-pregnant uterus. 

Although the transcription factor Nfil3 is essential for normal placental and embryonic development, it is not required for uNK cell development [199]. Moreover, assessing the gestational day 10 implantation sites showed that the deletion of T-bet did not modify the differentiation or function of uNK cells, while Eomes may regulate activation and IFN-γ production in uNK cells [200]. uNK cells may also play an essential role in the feto-maternal interface [201,202,203,204]. Vento et al. analyzed the transcriptome of about 70,000 single-cells from human first-trimester placentas, and they found three subsets decidual NK cells (dNK1. dNK2, and dNK3) [205]. These subsets co-express the tissue-resident markers CD49a (ITGA1) and CD9. dNK1 cells express CD39 (ENTPD1), CYP26A1, and B4GALNT1, whereas the defining markers of dNK2 cells are ANXA1 and ITGB2; dNK3 cells express CD160, KLRB1, and CD103 (ITGAE), but not the innate lymphocyte cell marker CD127 [205]. 

## 8. Clinical Relevance of trNK Cells

The role of cNK cells in tumor clearance is well established [64,206,207,208]. However, the anti-tumor functions of trNK cells are far from defined. Impairment in overall NK cell functions leads to an increased risk of cancer [209]. Tumor-infiltrating NK cells (TINKs) or tumor-associated NK cells (TANKs) [210] are involved in both clearances of malignancies and immunosuppression, depending on the tumor microenvironment (TMEs). However, the identities of the TINKs and TANKs within the TME are yet to be established. NK cells are involved in the clearance of hepatocellular carcinomas [211,212], colorectal carcinoma [213], gastric carcinoma [214], and squamous cell lung cancer [215]. Therapies formulated with haploidentical or autologous cNK cells hold promise to treat hematological malignancies, including acute myeloid leukemia (AML) and acute lymphoblastic leukemia (ALL) [216,217,218]. Moreover, allogeneic cNK cell-based therapies pose no or less risk of graft-versus-host disease (GvHD) [219]. Based on these observations, multiple Food and Drug Administration (FDA)-approved clinical trials using donor-derived cNK cells are being formulated and employed, including the ones expressing CAR [212,220,221,222,223,224]. Irrespective of these impressive advancements, the functional characterization, and utilization of cNK cells from human PBMCs, we are only beginning to understand the clinical relevance of trNK cells. 

Thymic NK cells are predicted to possess intrathymic functions and other functions in the secondary lymphoid organs or tissues when seeded extrathymically [102]. In mice, the Gata-3/IL-7-dependent thymic NK cells express a lesser level of inhibitory Ly49 receptors. Similar observations in humans are yet to be made. However, thymic NK cells are predicted to provide a regulatory function during the T cell ontogeny in the thymus, quite possibly in the elimination of negatively selected T cells. The CD127^+^ thymus-derived NK cells in mice (and CD56^bright^CD16^−^ in humans) in the LN produce a higher amount of inflammatory cytokines such as IFN-γ; however, they are also known to mediate a sub-optimal level of cytotoxicity compared to that of bone marrow-derived splenic NK cells [102]. Thus, the thymic NK cells have the potentials in regulating the expression levels of MHC class I on stromal cells. The clinical relevance of these findings is manifold. If the mechanism of how NK cells influence the T cell selection, it can be employed in methodologies where a T cell-based cellular therapy is being developed. It is not known whether at least some of the T cell-based autoimmune diseases are due to a defect in thymic NK cells.

Hepatic NK cells contain both the cNK and trNK cells. The number of these two subsets constitutes 50% of total hepatic lymphocytes [138]. Liver-specific CD56^bright^/CCR5^+^/CXCR6^+^/CD69^+^ trNK cells represent half of all NK cells in this organ, which are retained by the expression of homing chemokine receptors CXCR3, CXCR6, and CCR5 [34]. Given that the major function of the liver is detoxification, the trNK cells are of high clinical relevance. One of the major functions of trNK cells is the maintenance of hepatic tolerance and homeostasis. Interaction of trNK cells with hepatocytes via CD94/NKG2A-MHC non-classical class I-HLA-E results in the production of immunosuppressive factor TGF-β, which along with IL-6 from myeloid cells such a DCs, expand CD4^+^CD25^+^ regulatory T cells. Hepatocellular carcinoma is the leading liver-related malignancy, primarily caused by Hepatitis B or Hepatitis C viral infections. Acute hepatitis infection in humans leads to an altered trNK cell phenotype in the liver. Viral infections augment the expression of NCR1 (Nkp46) with an increased cytotoxic degranulation and inflammatory cytokine production [225]. Hepatic carcinomas and metastatic colorectal cancer in the liver contain a high number of intra-tumoral NCR1^+^ NK cells [226]. These observations emphasize the need to characterize and identify methods to augment the effector functions of tumor-infiltrating NK cells in the liver.

More than 80% of the lung NK cells are terminally-differentiated CD56^dim^CD16^+^ cells. The remaining 20% of the NK cells are composed of immature CD56^bright^CD16^−^ and less-differentiated CD56^dim^CD16^−^ cells. Lung-resident NK cells primarily identified using CD49a, CD69, and CD103 [227]. The upper respiratory system is the primary site of many viral infections, including influenza and Sars-Cov-2. The lung-resident CD56^bright^CD49a^+^ NK cells play an essential role in clearing influenza-infected epithelial cells and produce IFN-γ to facilitate the generation of CD8^+^ T-cell-based Tc1 or CD4^+^ T-cell-based Th1 responses. Moreover, the regeneration of tracheal epithelial cells depends on IL-22 produced by a subset of innate-like NK cells present in the lung [228,229]. In mice, clearance of pulmonary pseudometastases following the challenge with B16F10 melanoma strictly depends on the optimal functions of NK cells [230,231,232]. In patients with non-small-cell lung carcinoma (NSCLC), the number of tumor-infiltrating NK cells were significantly increased, which correlated with the overall survival rate [215]. Irrespective of these findings, the independent clinical relevance of lung-resident versus circulating NK cells is yet to be determined.

LN NK cells are a mixed population of circulating and thymus-derived NK cells [89,102]. Apart from this, a unique LN-derived NK subset has been recently identified [184]. NK cells in LNs mediate the interaction between innate and adaptive immune cells. LN is the major draining site where tumor- or pathogen-derived antigens are encountered by immune cells. NK cells in the LN are known to interact with antigen-bearing DCs and drive the differentiation of CD4^+^ T cells to induce early resistance to *Leishmania major* and *Toxoplasma gondii* [233,234]. Circulating NK cells are recruited in a CXCR3-dependent manner to LNs. IFN-γ production by these NK cells has an essential role in T_H_1 polarization [235]. A gradient of sphingosine-1 phosphate (S1P) in LNs position NK cells and regulate their IFN-γ responses [236]. Tumor-draining LNs are the first site of metastasis in most types of cancers and often used in staging cancer progression [237,238]. Using NK cells with TRAIL liposomes enhances their retention time within the tumor-draining LNs to induce apoptosis in cancer cells [237].

Uterine NK cells possess specific roles during pregnancy, including the formation of the fetal-maternal interface and placental vascular remodeling [239]. In humans, the dilation of uterine spiral arteries is attributed to trophoblasts. However, in mice, losing smooth muscles and dilation of vessels considered to be the functions of uNK cells [239]. NK cell-deficient mice have a defect in spiral artery remodeling [239]. Another role of uNK cells during pregnancy is to promote fetal development [120]. CD49a^+^Eomes^+^ trNK cells in the uterus have been defined, which secretes growth-promoting factors (GPF), including pleiotrophin and osteoglycin [240]. These CD49a^+^Eomes^+^ uNK cells enhance fetal growth during the early stages [240]. Decrease in the GPF-secreting NK cell subset impaired fetal development, resulting in fetal growth restriction [240]. Detailed studies are needed to further define uNK cell functions in the uterus to prevent the risk of pre-eclampsia. 

## 9. Summary and Future Outlook

The ability of NK cells to utilize germline-encoded non-clonotypic receptors to recognize and clear malignant or infected cells provide clinical promise. However, this paradigm of simplicity is challenged by the fact that NK cells are highly heterogeneous. This is further complicated by the findings that there are subsets of NK cells that are tissue-resident. These trNK cells display unique tissue-specific markers and are developmentally less mature [104]. These differences likely due to their local microenvironment and tissue localization. Despite recent insights that have helped to characterize trNK cells, essential questions remain unanswered. The term ‘tissue-resident’ is not fully defined at the transcriptional or transcriptomic levels. The cell plasticity is being tested in multiple models to differentiate the circulating versus the permanently tissue-residing NK cells. Determining their unique functions will vastly help in the formulations of specific NK cell-based therapies. Significant work needs to be performed in order to characterize trNK cells fully.

## Figures and Tables

**Figure 1 cancers-12-01553-f001:**
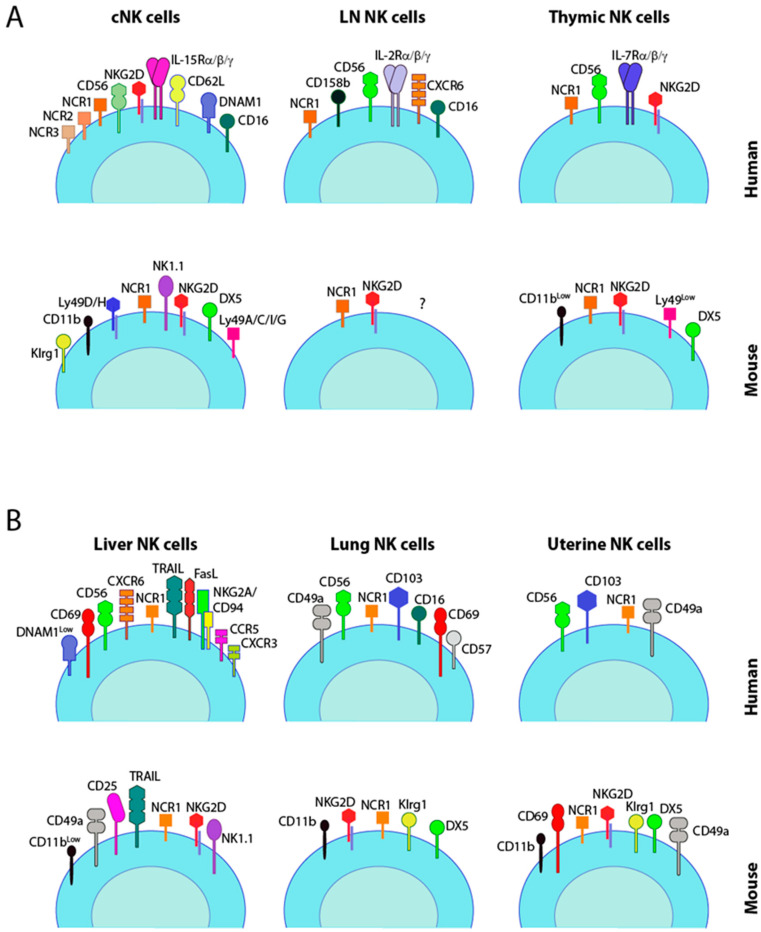
Phenotypic delineation of tissue-resident natural killer (trNK) cell subsets. Both human and mouse NK cells are shown. The surface markers of cNK cells from human blood or mice spleen are compared to the trNK cells from indicated tissues. (**A**) Conventional NK cells (cNK) are compared to lymph node (LN) and thymic NK cells. (**B**) Phenotypic markers of tissue-resident NK cells from the liver, lung, and uterus in humans and mice. CD56 is depicted in either bright green or light green to indicate the predominant presence of CD56^bright^ or CD56^dim^ subsets in the circulating conventional (cNK cells) or individual organs.

**Figure 2 cancers-12-01553-f002:**
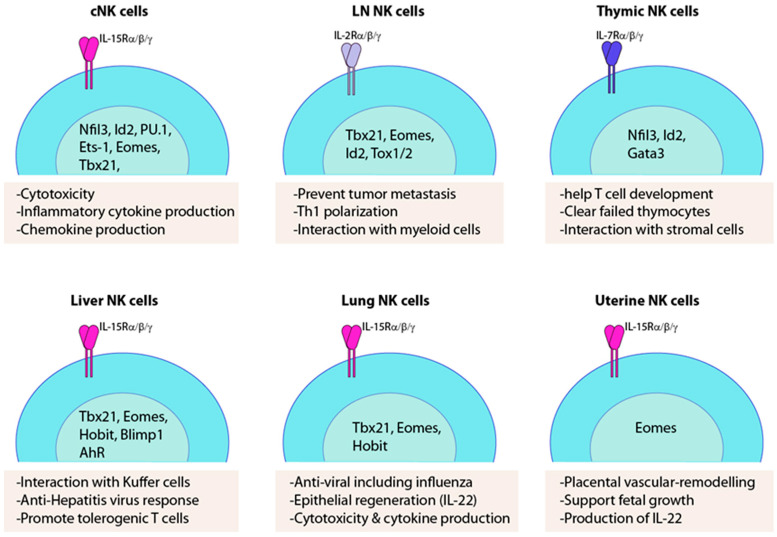
Unique transcription factor requirements and functions of trNK cells. Major common gamma chain receptor involved in the development of tissue-specific NK cells are indicated. Essential transcription factors that have been demonstrated are shown. A few of the defined functions of the trNK cells are listed under each subset.

**Figure 3 cancers-12-01553-f003:**
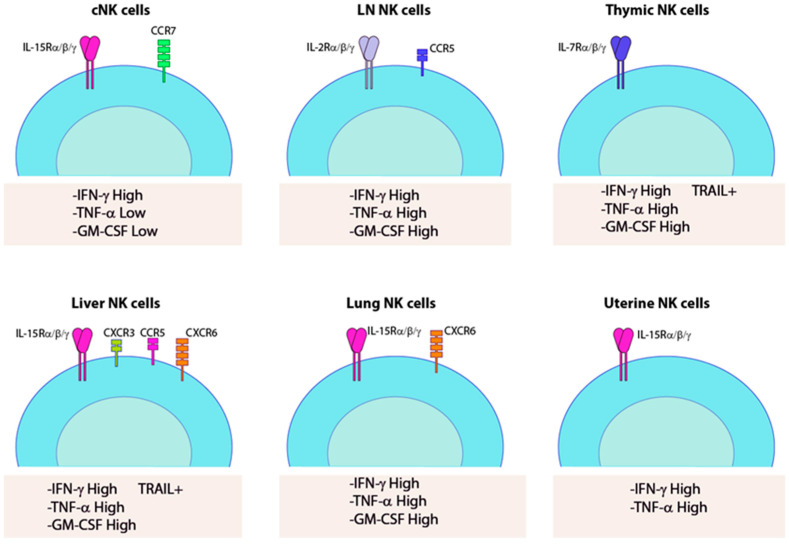
Inflammatory cytokine production from trNK cells. Cytokines produced from each type of trNK cells are shown along with the major cytokine and chemokine receptors.

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
