# Peer review of "Tissue-Resident NK Cells: Development, Maturation, and Clinical Relevance"

_cancers, 2020, doi:10.3390/cancers12061553_

Round 1

Reviewer 1 Report

Hashemi et al.   summarized the complex scenario of the cellular, molecular (transcription factors) phenotypic signatures (surface antigens), and divergent functions characterizing tissue-resident Vs conventional NKs. This is a very useful review, especially to those approaching to the NK cell biology and that are interested in the complex NK cell heterogeneity.

I have a concern on the “clinical relevance” word combination in the title since the pathological aspects of tissue-resident NK cells and the subsequent phenotype/functional alterations should be deepened in the review.

In line with this comment on the title, I would have expected a section on the clinical relevance, while authors concluded the review with a “Summary and future outlook” section, that appears a too fast conclusion. Why authors do not discuss the clinical relevance of the knowledge generated by the differences between conventional and resident NK cells? What about the possibility to convert, by different approaches, the resident NK cells to conventional NK cells and vice versa?

There are several works (10.1038/nrc.2017.51, PMID:23441128, 10.1158/2159-8290.CD-15-0732) showing that NK cells infiltrating different tumour tissue acquire phenotype and functions of decidual/uterine NK cells. I would comment on this data, supporting the idea that NK cell phenotype/functions are strongly influenced by the host tissue and the physio/pathological conditions.

I’m not sure that the audience of Cancers journal might be totally confident with some of the acronym present in the review and strictly related to the immunology filed. I would suggest preparing a list of abbreviation, to make more comprehensive the text also to those not expert in immunology that would approach to NK cell biology.

Minor comments:

Page 2 line 66: I would replace  “fail to express normal levels of MHC-I” with “have null or low expression of MHC-I”

Page 2 line 67: here authors are mentioning on the regulatory activities of NK cells. I would not restrict this only to neutrophils and macrophages. NK cells also interact with MDSCs, DCs and T cells. I would stress also on this point.

Figures.

Figures provided a clear scenario of similarities and differences between conventional NK cells and tissue-resident NK cells, in a view of surface antigens and transcription factors. Therefore, as for macrophages and T cells, classification of different cell subsets includes also their cytokine repertoire that is profoundly changed, according to the tissue of residency and the physiological or pathological conditions. Therefore, I strongly recommend providing an additional Figure 3, that while maintaining the scheme of Figure 1 and Figure 2, shows also the different cytokine repertoire. This will significantly increase the quality of the work, by providing the complete view of similarities and differences between conventional NK cells and tissue-resident NK cells.

I will consider the review suitable for publication, pending the minor revisions suggested.

Author Response

Reviewer #1

Comments and Suggestions for Authors

Hashemi et al.   summarized the complex scenario of the cellular, molecular (transcription factors) phenotypic signatures (surface antigens), and divergent functions characterizing tissue-resident Vs conventional NKs. This is a very useful review, especially to those approaching to the NK cell biology and that are interested in the complex NK cell heterogeneity.

I have a concern on the “clinical relevance” word combination in the title since the pathological aspects of tissue-resident NK cells and the subsequent phenotype/functional alterations should be deepened in the review.

In line with this comment on the title, I would have expected a section on the clinical relevance, while authors concluded the review with a “Summary and future outlook” section, that appears a too fast conclusion. Why authors do not discuss the clinical relevance of the knowledge generated by the differences between conventional and resident NK cells? What about the possibility to convert, by different approaches, the resident NK cells to conventional NK cells and vice versa?

As requested, we have added a new section describing these. We thank the Reviewer for the suggestion.

There are several works (10.1038/nrc.2017.51, PMID:23441128, 10.1158/2159-8290.CD-15-0732) showing that NK cells infiltrating different tumour tissue acquire phenotype and functions of decidual/uterine NK cells. I would comment on this data, supporting the idea that NK cell phenotype/functions are strongly influenced by the host tissue and the physio/pathological conditions.

We thank the Reviewer. This is a good suggestion and we have included this information.

I’m not sure that the audience of Cancers journal might be totally confident with some of the acronym present in the review and strictly related to the immunology filed. I would suggest preparing a list of abbreviation, to make more comprehensive the text also to those not expert in immunology that would approach to NK cell biology.

We are providing list of abbreviations and some acronyms that will influence the ease of read of the paper.

Minor comments:

Page 2 line 66: I would replace  “fail to express normal levels of MHC-I” with “have null or low expression of MHC-I”.

Changes are made.

Page 2 line 67: here authors are mentioning on the regulatory activities of NK cells. I would not restrict this only to neutrophils and macrophages. NK cells also interact with MDSCs, DCs and T cells. I would stress also on this point.

We thank the Reviewer for the suggestion and have included this information in the current version of the manuscript.

Figures.

Figures provided a clear scenario of similarities and differences between conventional NK cells and tissue-resident NK cells, in a view of surface antigens and transcription factors. Therefore, as for macrophages and T cells, classification of different cell subsets includes also their cytokine repertoire that is profoundly changed, according to the tissue of residency and the physiological or pathological conditions. Therefore, I strongly recommend providing an additional Figure 3, that while maintaining the scheme of Figure 1 and Figure 2, shows also the different cytokine repertoire. This will significantly increase the quality of the work, by providing the complete view of similarities and differences between conventional NK cells and tissue-resident NK cells.

As suggested by the Reviewer, we have added a Figure 3 describing the cytokine repertoire of trNK cells.

I will consider the review suitable for publication, pending the minor revisions suggested.

Reviewer 2 Report

In this review article entitled “Tissue-Resident NK cells: Development, Maturation, and the Clinical Relevance” the authors describe the heterogeneity and functional diversity of tissue-resident NK cells. The review discusses an important topic. It will be a very precious addition to the literature.

I would like to suggest the authors briefly discuss the current evidence of human NK cells in tumor tissues of patients with hepatocellular carcinoma. This could add more values to this paper.

The following points should be addressed by the authors when revising their manuscript:

Figure 1, panel B: should be “CCR5” instead of “CXR5”.

Line 101: should be “endometrium” instead of “Endometrium”.

Line 167: should be “subset” instead of “sub-set”.

Line 189: TRAIL should be defined at line 144.

Line 217: there are different punctuation mistakes.

Line 223: should be “et al. ” instead of “et al”.

Line 224: please use consistent nomenclature (should be trNK instead of tissue-resident NK).

Line 228: please define the term “ TRM cells” here.

Line 239: should be “lung-resident subset” instead “Lung-resident subset”

Line 250: should be “cortex” instead of “Cortex”.

Line 262: please define the term “HEV”.

Line 266: please use consistent nomenclature (should be “(Ltr)” instead of “lymphoid tissue-resident”).

Line 274: please substitute “Caliguri” for “Freud et al.”.

Line 360: should be “liver” instead of “Liver”.

Reference 133: should be “Tissue-resident Eomes(hi) T-bet(lo) CD56(bright) NK cells with reduced proinflammatory potential are enriched in the adult human liver. European journal of immunology”.

Author Response

Reviewer #2

In this review article entitled “Tissue-Resident NK cells: Development, Maturation, and the Clinical Relevance” the authors describe the heterogeneity and functional diversity of tissue-resident NK cells. The review discusses an important topic. It will be a very precious addition to the literature.

I would like to suggest the authors briefly discuss the current evidence of human NK cells in tumor tissues of patients with hepatocellular carcinoma. This could add more values to this paper.

We thank the Reviewer and have added this information by introducing a new section.

The following points should be addressed by the authors when revising their manuscript:

Figure 1, panel B: should be “CCR5” instead of “CXR5”.

Thank you and we have changed it.

Line 101: should be “endometrium” instead of “Endometrium”.

Changes are made.

Line 167: should be “subset” instead of “sub-set

Changes are made.

Line 189: TRAIL should be defined at line 144.

Changes are made.

Line 217: there are different punctuation mistakes.

Changes are made.

Line 223: should be “et al. ” instead of “et al”.

Changes are made.

Line 224: please use consistent nomenclature (should be trNK instead of tissue-resident NK).

 Changes are made.

Line 228: please define the term “ TRM cells” here.

Changes are made.

Line 239: should be “lung-resident subset” instead “Lung-resident subset”

Changes are made.

Line 250: should be “cortex” instead of “Cortex”.

Changes are made.

Line 262: please define the term “HEV”.

Changes are made.

Line 266: please use consistent nomenclature (should be “(Ltr)” instead of “lymphoid tissue-resident”).

Changes are made.

Line 274: please substitute “Caliguri” for “Freud et al.”.

Changes are made.

Line 360: should be “liver” instead of “Liver”.

Changes are made.

Reference 133: should be “Tissue-resident Eomes(hi) T-bet(lo) CD56(bright) NK cells with reduced proinflammatory potential are enriched in the adult human liver. European journal of immunology”.

Changes are made.

Reviewer 3 Report

Hashemi and Malarkannan, Tissue-Resident NK cells: Development, Maturation, and the Clinical Relevance

The manuscript has an introduction to NK cells, a section comparing conventional NK cells to tissue resident NK cells, and then another five sections describing characteristics of thymic, liver-, lung, lymph-node resident NK cells, and uterine NK cells. In the end, the authors have a written a summary with future outlook. The manuscript includes two figures giving an overview of phenotypic markers and other characteristics of conventional NK cells compared to the five mentioned tissue-resident NK.

The manuscript is well organized and the authors refers to central publications in the research field. However, the manuscript suffers from grammatical/language issues and in some parts information is placed without good enough context for the readers.  

There is no comment on NK cells/tissue-residency in the gut or the reason why they are not included, which I think could have been addressed.

The intention of the figures are good but it is not clearly stated what they are showing and therefore confusing. The markers they have chosen to show in figure 1 seems a bit arbitrary and needs further explanation. The figure text need to be clearly improved and markers verified, or the figure should be removed. The authors is only describing surface receptors so the title should reflect that. Some of my initial questions: Are you comparing cNK from human blood and mice spleen? Would you say a CD56 bright NK cell is not an cNK? KIRs are only included in the lymph nodes? I.e. is the absence of CD49a and CD25 in human liver NK cells seems incorrect? (Martrus et al).

Figure text for figure 2 should also be elaborated and corrected for language

The last summary section seems especially unfinished with a many grammar and language issues and should be rewritten if to be published.

Some of the minor comments are listed below:

L 27: lymphoidal – lymphoid

L 46: as their development – during/through their development

L 55: encoding - encoded

L 56. Activation receptors – Activating receptors

L 57: Non-classical MHC class I - Non-classical MHC class I related

L 83-84: Unclear if this refers to cNK or trNK

L92: Besides what? Unclear use of besides (also other places in the text)

L 93: another example of what?

L 100: I think we can say NK have potent cytolytic capacity, not just potentially…

L 101: Outnumbered… and lowercase e in endometrium.

L 111: niche in an organ – organ specific niches

L 113: Incomplete sentence. Not clear what the authors mean

L 138: absent expression of Ly49D

L 144-145: Do the authors mean killing by the means of perforin etc? Unclear

L 155: lymphoidal – lymphoid

L 157-158: Reference to figure 1, but many of the markers mentioned are not even depicted in the figure. It is confusing

L 184: CD49a+DX5+ NK cells of host origin – should this be CD49a+DX5- NK cells of host origin?

Author Response

Reviewer #3

The manuscript has an introduction to NK cells, a section comparing conventional NK cells to tissue resident NK cells, and then another five sections describing characteristics of thymic, liver-, lung, lymph-node resident NK cells, and uterine NK cells. In the end, the authors have a written a summary with future outlook. The manuscript includes two figures giving an overview of phenotypic markers and other characteristics of conventional NK cells compared to the five mentioned tissue-resident NK.

The manuscript is well organized and the authors refers to central publications in the research field. However, the manuscript suffers from grammatical/language issues and in some parts information is placed without good enough context for the readers.  

We apologize for our omissions. We have made significant amount changes.

There is no comment on NK cells/tissue-residency in the gut or the reason why they are not included, which I think could have been addressed.

We have included some information related to the NK cells present in the gut. Hope this sufficient ground for the reader.

The intention of the figures are good but it is not clearly stated what they are showing and therefore confusing. The markers they have chosen to show in figure 1 seems a bit arbitrary and needs further explanation. The figure text need to be clearly improved and markers verified, or the figure should be removed. The authors is only describing surface receptors so the title should reflect that. Some of my initial questions: Are you comparing cNK from human blood and mice spleen?

Yes, We correct it in the figure caption. Line 381-382

 Would you say a CD56 bright NK cell is not an cNK?

We have modified the figure legend to indicate the status of the CD56 expression.

KIRs are only included in the lymph nodes? I.e. is the absence of CD49a and CD25 in human liver NK cells seems incorrect? (Martrus et al).

We provide these markers based on the published data, some markers like CD25 on mice or KIRs on other tissue resident than LNs are not depicted, as we did not find related information and it does not mean they may not exist on the cells.

Figure text for figure 2 should also be elaborated and corrected for language

Changes are made.

The last summary section seems especially unfinished with a many grammar and language issues and should be rewritten if to be published.

We apologize for this omission and we have rewritten the 'Summary and the future outlook'.

Some of the minor comments are listed below:

L 27: lymphoidal – lymphoid

Changes are made.

L 46: as their development – during/through their development

Changes are made.

L 55: encoding – encoded

Changes are made.

L 56. Activation receptors – Activating receptors

Changes are made.

L 57: Non-classical MHC class I - Non-classical MHC class I related

Changes are made.

L 83-84: Unclear if this refers to cNK or trNK

Changes are made. This refers to cNK cells

L92: Besides what? Unclear use of besides (also other places in the text)

Changes are made. We have changed ‘Besides’ to ‘Also’, which make more grammatical sense.

L 93: another example of what?

We have removed ‘another example’.

L 100: I think we can say NK have potent cytolytic capacity, not just potentially…

Changes are made. We have removed ‘potentially’.

L 101: Outnumbered… and lowercase e in endometrium.

Changes are made.

L 111: niche in an organ – organ specific niches

Changes are made.

L 113: Incomplete sentence. Not clear what the authors mean

Changes are made.

L 138: absent expression of Ly49D

Changes are made.

L 144-145: Do the authors mean killing by the means of perforin etc? Unclear

Changes are made.

L 155: lymphoidal – lymphoid

Changes are made.

L 157-158: Reference to figure 1, but many of the markers mentioned are not even depicted in the figure. It is confusing.

We selected the essential markers to show in the figure.

L 184: CD49a+DX5+ NK cells of host origin – should this be CD49a+DX5- NK cells of host origin?

Changes are made.

Reviewer 4 Report

This is a review article of an interesting and relevant subject. There is a lot of detailed information which is well referenced, and the types of NK cells discussed in subheadings is a nice way to review the subject. Unfortunately I found the review hard to read and it did not flow well. In general I found the whole article clunky and lacked flow, often introducing statements came after the specific information, and in places a lack of scientific language. Please find some details of lines with grammar errors and some specific comments below:

Introduction

The information is mostly correct however it does feel like more a list of things NK cells express and do, rather than a review introduction/ setting the scene for the more in depth understanding of tissue resident cells.

e.g. Line 66 NK cells participate in recruiting other innate immune cells, such as neutrophils and macrophages (81, 82).   Perhaps explain why/how/why this is relevant.

Line 27 – I think the first and possibly second lines are not necessary and complicate the introduction. Could simply begin… At the site of pathology migrated lymphocytes…

Line 46 grammar

Line 50 Expression of IL-2b receptor (CD22) makes an irreversible lineage commitment to NK cells (40, 41, 44) . I wouldnt use ‘make’ to describe lineage commitment

Line 49. Expression of IL-2b receptor (CD22) makes an irreversible lineage commitment to NK cells (40, 41, 44). Immature NK cells express CD117 and integrin a2 (DX5) (45). In the next stage, these cells downregulate CD27 and upregulate CD11b (46).

The section above is describing key NK cells markers and NK cell development, you could condense or have a table. The sentences don’t flow well and there are detailed reviews to reference here e.g. Yu et al Trends in Immunol 2013.

Line 56 references from this sentence are about the receptors themselves and not specifically about the ligands. Relevant references and rewriting of this sentence is needed.

Line 65 grammar

Lines 96 and 97  No references and no information about relevance. These are a very interesting points but it means nothing without data or thinking to back to up. This point is discussed later in the review but I feel needs some detail here.

Thymic NK cells.

 This section felt muddled and was very difficult to follow.

Line 214 I would argue all organs are directly ‘prone’, clarify.

Line 294 define non menstrual cycle, confusing terminology.

References 194 and 195 I think have got mixed up or are incorrect.

 Summary is not well written and content too hard to follow

Author Response

Reviewer #4

This is a review article of an interesting and relevant subject. There is a lot of detailed information which is well referenced, and the types of NK cells discussed in subheadings is a nice way to review the subject. Unfortunately I found the review hard to read and it did not flow well. In general I found the whole article clunky and lacked flow, often introducing statements came after the specific information, and in places a lack of scientific language. Please find some details of lines with grammar errors and some specific comments below:

Introduction

The information is mostly correct however it does feel like more a list of things NK cells express and do, rather than a review introduction/ setting the scene for the more in depth understanding of tissue resident cells.

We appreciate your comment. We have modified and improved the text for flow and content.

e.g. Line 66 NK cells participate in recruiting other innate immune cells, such as neutrophils and macrophages (81, 82).   Perhaps explain why/how/why this is relevant.

Thank you again. We have improved the contents in the current version of the manuscript.

Line 27 – I think the first and possibly second lines are not necessary and complicate the introduction. Could simply begin… At the site of pathology migrated lymphocytes…

Thank you and we have simplified these sentences.

Line 46 grammar

Changes are made.

Line 50 Expression of IL-2b receptor (CD22) makes an irreversible lineage commitment to NK cells (40, 41, 44) . I wouldnt use ‘make’ to describe lineage commitment

Changes are made.

Line 49. Expression of IL-2b receptor (CD22) makes an irreversible lineage commitment to NK cells (40, 41, 44). Immature NK cells express CD117 and integrin a2 (DX5) (45). In the next stage, these cells downregulate CD27 and upregulate CD11b (46).

The section above is describing key NK cells markers and NK cell development, you could condense or have a table. The sentences don’t flow well and there are detailed reviews to reference here e.g. Yu et al Trends in Immunol 2013.

Thank you and we have simplified this section and included references for additional reading.

Line 56 references from this sentence are about the receptors themselves and not specifically about the ligands. Relevant references and rewriting of this sentence is needed.

Changes are made.

Line 65 grammar

Changes are made.

Lines 96 and 97  No references and no information about relevance. These are a very interesting points but it means nothing without data or thinking to back to up. This point is discussed later in the review but I feel needs some detail here.

Changes are made.

Thymic NK cells.

 This section felt muddled and was very difficult to follow.

Thank you and we have modified the text and have improved this section.

Line 214 I would argue all organs are directly ‘prone’, clarify.

This sentence is removed in the current version of the manuscript.

Line 294 define non menstrual cycle, confusing terminology.

Changes are made.

References 194 and 195 I think have got mixed up or are incorrect.

Thank you. References are fixed.

 Summary is not well written and content too hard to follow

Changes are made.

Round 2

Reviewer 3 Report

The authors have done some corrections and added new text to accommodate referee comments. In my opinion, the manuscript has unfortunately not been improved to such a standard that I will recommend publication.

It strikes me when reading it again that the title:  “Tissue-Resident NK cells: Development, Maturation, and the Clinical Relevance” is maybe a bit premature. As the authors state from line 79, “the developmental stages of trNK cells are poorly defined”, “At what stage the commitment to trNK cells happen, is unknown”, “Moreover, whether specific cytokines drive NK cell progenitors toward tissue residency is an open question”. “To what extend cytokine profiles of trNK cells differ from cNK cells is still under investigation”. As the authors states themselves in the abstract, this review is more focused on the phenotypic signatures and functions of trNK, and there is not so much information about development and maturation.

The introduction has in the second revision been extended with a long section of how conventional NK cells interacts with DCs. I don’t believe the authors are successful in putting this into context of trNKs. 

Line 82-86: The description of the effect of cytokines on nutrient transporters without any other introduction or discussion seems a bit out of context.

Section 6 is very confusing. It is not clear whether the authors are talking about cNK or trNK.

Some of the grammatical/language issues were corrected. Unfortunately, a long range of new ones were introduced.  The reader gets the impression that the manuscript are written in a hurry and some places it does not flow well.  

Some of the typos and language issues are listed below:

Line 69: title have double trNK

Line 82: extent

Line 83: are

Line 85 compared

Figure 1 text in line 90: extra text

Line 91: ‘that’ should be removed

LILne 101: remove comma after TNF-a

Line 110: The latter subset…

Line 112: ‘and’ between niche and transcriptional networks?

Figure 2, line 118. Few of the defined functions... Should it rather be: ‘A few of the defined functions….

Line 120: The humans?

Line 136-137: several language issues

Line 196: The trNK are CD49+DX5-, not CD49+DX5+?

Line 227: Language ...’each trNK cells’…Rewrite to ‘each type of trNK cell’?

Line 233: compared

Line 235-236: language

Line 242. Grammar

Line 245: what does ‘These’ refer to?

Line 252-253: language

Line 261: ‘The’ human body…

Line 263: language

Line 273-274: language

Line 279: Rewrite? In the lymph node, 60% of total NK cells are … Ltr-NK?

Line 309-310: language? Unclear.

Line 329: ..subsets of decidual

Line 351: What does ‘it’ refer to?

Line 354: Nomenclature: use CD56 bright as in the rest of the manuscript

Lines 357 to 408 have many language problems and need more work. It does not flow well. Line 366-371 is short on references.

Line 406: grammar

Line 417: something is missing before ‘phenotypically’

Author Response

The authors have done some corrections and added new text to accommodate referee comments. In my opinion, the manuscript has unfortunately not been improved to such a standard that I will recommend publication.

It strikes me when reading it again that the title:  “Tissue-Resident NK cells: Development, Maturation, and the Clinical Relevance” is maybe a bit premature. As the authors state from line 79, “the developmental stages of trNK cells are poorly defined”, “At what stage the commitment to trNK cells happen, is unknown”, “Moreover, whether specific cytokines drive NK cell progenitors toward tissue residency is an open question”. “To what extend cytokine profiles of trNK cells differ from cNK cells is still under investigation”. As the authors states themselves in the abstract, this review is more focused on the phenotypic signatures and functions of trNK, and there is not so much information about development and maturation.

We thank the Reviewer for all the comments to help to improve our manuscript. We have extensively modified the manuscript and hope the current version is acceptable.

The introduction has in the second revision been extended with a long section of how conventional NK cells interacts with DCs. I don’t believe the authors are successful in putting this into context of trNKs. 

Again, we have extensively modified the Introduction based on the comments provided by the Reviewer.

Line 82-86: The description of the effect of cytokines on nutrient transporters without any other introduction or discussion seems a bit out of context.

We thank the Reviewer and we have made changes in this section to make it more meaningful and contextual.

Section 6 is very confusing. It is not clear whether the authors are talking about cNK or trNK.

We have made appropriate changes to avoid confusions.

Some of the grammatical/language issues were corrected. Unfortunately, a long range of new ones were introduced.  The reader gets the impression that the manuscript are written in a hurry and some places it does not flow well.  

Some of the typos and language issues are listed below:

We have modified all these and additional grammatical errors in the current version of the manuscript.

Line 69: title have double trNK

Line 82: extent

Line 83: are

Line 85 compared

Figure 1 text in line 90: extra text

Line 91: ‘that’ should be removed

LILne 101: remove comma after TNF-a

Line 110: The latter subset…

Line 112: ‘and’ between niche and transcriptional networks?

Figure 2, line 118. Few of the defined functions... Should it rather be: ‘A few of the defined functions….

Line 120: The humans?

Line 136-137: several language issues

Line 196: The trNK are CD49+DX5-, not CD49+DX5+?

Line 227: Language ...’each trNK cells’…Rewrite to ‘each type of trNK cell’?

Line 233: compared

Line 235-236: language

Line 242. Grammar

Line 245: what does ‘These’ refer to?

Line 252-253: language

Line 261: ‘The’ human body…

Line 263: language

Line 273-274: language

Line 279: Rewrite? In the lymph node, 60% of total NK cells are … Ltr-NK?

Line 309-310: language? Unclear.

Line 329: ..subsets of decidual

Line 351: What does ‘it’ refer to?

Line 354: Nomenclature: use CD56 bright as in the rest of the manuscript

Lines 357 to 408 have many language problems and need more work. It does not flow well. Line 366-371 is short on references.

Line 406: grammar

Line 417: something is missing before ‘phenotypically’

Reviewer 4 Report

This version of the manuscript represents a significant improvement in both style and content. There are some minor language and grammar corrections that need attention (I have document some, attached). The section on clinical use of NK cells appears much longer and is comprehensive with respective to role of trNK, however I am not certain their clinical relevance comes across.

64 interact

69 trNK long hand first

74 double space

99 XLCL1 ?

120 In humans

141 ‘has been’ not ‘is’

147 are yet

261 The human body has

274 is yet

357 is manifold

363 are retained

395 in a CXCR3

440 spiral…spinal maybe ?

Author Response

This version of the manuscript represents a significant improvement in both style and content. There are some minor language and grammar corrections that need attention (I have document some, attached). The section on clinical use of NK cells appears much longer and is comprehensive with respective to role of trNK, however I am not certain their clinical relevance comes across.

We thank the Reviewer and have changes all the following and additional grammatical errors in the current version of the manusctipt.

64 interact

69 trNK long hand first

74 double space

99 XLCL1 ?

120 In humans

141 ‘has been’ not ‘is’

147 are yet

261 The human body has

274 is yet

357 is manifold

363 are retained

395 in a CXCR3

440 spiral…spinal maybe ?

Round 3

Reviewer 3 Report

The manuscript has approved significantly. Section 6 is now much clearer and the authors have given context to statements that previously seemed misplaced. Most of the languages issues are corrected and it is easier to read. I recommend publication of the manuscript in the present form.